# Towards Climate Resilient and Environmentally Sustainable Health Care Facilities

**DOI:** 10.3390/ijerph17238849

**Published:** 2020-11-28

**Authors:** Carlos Corvalan, Elena Villalobos Prats, Aderita Sena, Diarmid Campbell-Lendrum, Josh Karliner, Antonella Risso, Susan Wilburn, Scott Slotterback, Megha Rathi, Ruth Stringer, Peter Berry, Sally Edwards, Paddy Enright, Arabella Hayter, Guy Howard, Jostacio Lapitan, Margaret Montgomery, Annette Prüss-Ustün, Linda Varangu, Salvatore Vinci

**Affiliations:** 1School of Public Health, University of Sydney, Camperdown, NSW 2006, Australia; 2Department of Environment, Climate Change and Health, World Health Organization, 1211 Geneva, Switzerland; asena@who.int (A.S.); campbelllendrumd@who.int (D.C.-L.); haytera@who.int (A.H.); MontgomeryM@who.int (M.M.); pruessa@who.int (A.P.-U.); vincis@who.int (S.V.); 3Health Care without Harm, Reston, VA 20190, USA; josh@hcwh.org (J.K.); antonella@hcwh.org (A.R.); swilburn@hcwh.org (S.W.); mrathi@hcwh.org (M.R.); rstringer@hcwh.org (R.S.); 4Independent Consultant, San Francisco, CA 94117, USA; scott.slotterback@gmail.com; 5Department of Geography and Environmental Management, University of Waterloo, Waterloo, ON N2L 3G1, Canada; peter.berry@canada.ca (P.B.); pmenright@uwaterloo.ca (P.E.); 6Department of Communicable Diseases and Environmental Determinants of Health, Pan American Health Organization, Washington, DC 20037, USA; edwardss@paho.org; 7Civil Engineering, Queens Building, University Walk, Bristol BS8 1TR, UK; guy.howard@bristol.ac.uk; 8Department of Health Security Preparedness, World Health Organization, 1211 Geneva, Switzerland; lapitanj@who.int; 9Canadian Coalition for Green Health Care, Halifax, NS B3J 1Z4, Canada; Linda@greenhealthcare.ca

**Keywords:** climate resilience, environmental sustainability, climate change and health, health care facilities

## Abstract

The aim of building climate resilient and environmentally sustainable health care facilities is: (a) to enhance their capacity to protect and improve the health of their target communities in an unstable and changing climate; and (b) to empower them to optimize the use of resources and minimize the release of pollutants and waste into the environment. Such health care facilities contribute to high quality of care and accessibility of services and, by helping reduce facility costs, also ensure better affordability. They are an important component of universal health coverage. Action is needed in at least four areas which are fundamental requirements for providing safe and quality care: having adequate numbers of skilled human resources, with decent working conditions, empowered and informed to respond to these environmental challenges; sustainable and safe management of water, sanitation and health care waste; sustainable energy services; and appropriate infrastructure and technologies, including all the operations that allow for the efficient functioning of a health care facility. Importantly, this work contributes to promoting actions to ensure that health care facilities are constantly and increasingly strengthened and continue to be efficient and responsive to improve health and contribute to reducing inequities and vulnerability within their local settings. To this end, we propose a framework to respond to these challenges.

## 1. Introduction

As the climate continues to change, risks to health systems and facilities—including hospitals, clinics and community care centers—are increasing, reducing the ability of health professionals to protect people from a range of climate hazards. Health care facilities (HCFs) are the first and last line of defense to climate change impacts. They can be responsible for large emissions of greenhouse gases (GHGs), but also they provide the needed services and care to people harmed by extreme weather and other climate hazards. HCFs can also produce large amounts of environmental waste and contamination (GHGs and other contaminants) which may be infectious, toxic or radioactive and therefore a threat to the health of individuals and communities. HCFs provide health treatments and related procedures to patients and vary in size from small health care clinics to very large hospitals. In many countries, they often lack functioning infrastructure, an informed and trained health workforce to address environmental challenges, and are subject to inadequate energy supplies, water, sanitation and waste management services. Improving these is a priority and is key to building resilience and contributing to environmental sustainability.

It is therefore urgent to work to enhance the capacity of HCFs to protect and improve the health of their target communities in an unstable and changing climate and to empower HCFs to be environmentally sustainable, by optimizing the use of resources and minimizing the release of waste into the environment. Climate resilient and environmentally sustainable HCFs contribute to a high quality of care and accessibility of services, and by helping reduce facility costs, also contribute to better affordability. They are, therefore, an important component of universal health coverage (UHC). In 2015, the World Health Organization (WHO) released the Operational Framework for Building Climate Resilient Health Systems (Operational Framework) [1], proposing ten areas of interventions. Here, we focus on four, which are fundamental requirements for providing safe and quality care: (i) adequate numbers of skilled human resources with decent working conditions, empowered and informed to respond to environmental challenges; (ii) sustainable and safe management of water, sanitation and health care waste services; (iii) sustainable energy services; (iv) appropriate infrastructure, technologies, products and processes, including all the operations that allow for the efficient functioning of the HCF.

Although there is much that can be accomplished by the actions within HCFs, effective interventions towards strengthening climate resilience and environmental sustainability often depend on good cross-sectoral action. This is particularly true for water and energy access, construction, building retrofitting, treatment and removal of health care waste, environmental standards, supply chains, timely information and surveillance. Thus, many actions need to be undertaken by sectors and decision makers outside the HCF and therefore health sector officials will need to influence, inform and request interventions by local and national governments and policymakers.

## 2. Understanding Climate Resilience and Environmental Sustainability of HCFs

There are several definitions that support our understanding of these subjects. Health systems include an ensemble of all public and private organizations, institutions and resources mandated to improve, maintain or restore health as well as incorporate disease prevention, health promotion and efforts to influence other sectors to address health concerns in their policies [2]. Health system resilience is the capacity of health actors, institutions and populations to prepare for and effectively respond to crises; maintain core functions when a crisis hits; as well as stay informed through lessons learned during the crisis and reorganize if conditions require it [3]. It is the ability to absorb disturbance, to adapt and to respond with the provision of needed services [4].

Health care facilities are settings that provide direct health treatment procedures for patients and include hospitals and health care clinics. In the context of emergencies, HCFs are hospitals, primary health care centers, isolation camps, feeding centers and others [5]. We understand resilience, in the context of climate change, as the capacity of social, economic and environmental systems to cope with a hazardous event, trend or disturbance, responding or reorganizing in ways that maintain their essential function, identity and structure while also maintaining the capacity for adaptation, learning and transformation [6].

### 2.1. What Do We Understand by Climate Resilience in HCFs?

Climate resilient HCFs are those able to anticipate, respond to, cope with, recover from and adapt to climate-related shocks and stress, so as to bring ongoing and sustained health care to their target populations, despite an unstable climate [1]. Figure 1 illustrates the important dynamics affecting the climate resilience of HCFs. Building on the concept of risk as a function of hazards, vulnerabilities and exposures [6,7] (illustrated in the figure as a triangle, as defined by the Intergovernmental Panel on Climate Change (IPCC)), it depicts how hazards, in the form of a sudden event (a shock, such as a storm or sudden flood), or a slow-onset event (a stress, such as a drought, sea-level rise or high volume of cases of a climate-related disease), will reduce the HCFs’ level of performance and capacity (left axis). This would occur through a combination of impacts on key facility elements (for example, increasing—or adding to—the vulnerability of the health workforce, its infrastructure, its water, sanitation and energy systems), and therefore increasing risks. The level of resilience (right axis) indicates whether the facility will recover its pre-event state, recover but to a state worse than before (or even collapse and not recover) or recover and attain a level of resilience greater than before the event. The figure also highlights the risk management steps for prevention, preparedness, response and recovery [8].

Key interventions to build resilience in HCFs include bolstering the health workforce (such as training, communications), optimizing access to food, water, sanitation and health care waste services through monitoring, assessment and management and improving access and reliability of energy sources (such as back-up systems, alternative sources of energy, emergency plans), as well as the adaptation of infrastructures and technologies (such as building retrofits, adoption of new systems and technologies, sustainability of operations). The WHO Operational Framework provides additional areas to be considered in specific contexts for strengthening climate resilience in HCFs, such as strengthening health information systems, or performing climate change, health vulnerability and adaptation assessments [1], which can provide the necessary information and stakeholder partnerships to support these activities.

### 2.2. What Do We Understand by Environmental Sustainability in HCFs?

Environmentally sustainable health care facilities are those that improve, maintain or restore health, while minimizing negative impacts on the environment and leveraging opportunities to restore and improve it [10]. Following the concept of risk as defined by the IPCC [6,7], environmental sustainability aims to reduce hazards resulting from HCF operations (such as health care waste), while simultaneously working towards decreasing exposures and vulnerabilities (both within and outside the HCF) (Figure 2). The IPCC notes with very high confidence that the most effective vulnerability reduction measures for health in the near term are programs that implement and improve basic public health measures, such as the provision of clean water and sanitation, secure essential health care, including vaccination and child health services, increase capacity for disaster preparedness and response and alleviate poverty [6].

Facilities need to also optimize their use of natural resources, principally that of water and energy, ensuring a balance that is not too low to maintain good functioning, nor too high to waste and deplete resources. Thus, in many HCFs in low-resource settings, the aim is to increase their access and use of water and energy. Interventions for environmental sustainability are key to moving from higher risk (left side of the graphic in Figure 2) to lower risk situations (right side of the graphic in Figure 2). Examples of hazards that threaten environmental sustainability include biological hazards (epidemics, pests) and human-made hazards (chemical, radiological, biological wastes, water and energy supply disruptions, air pollution, food and water contamination, insufficient health workforce). Interventions to reduce hazards occur mostly in HCFs (lower half of the graphic in Figure 2), while interventions to reduce vulnerability and exposure occur mostly in patients, the health workforce and the wider community (upper half of the graphic in Figure 2).

## 3. Strengthening Climate Resilience and Environmental Sustainability

HCFs and, more broadly, the health sector, though profoundly impacted by climate-related shocks and stresses, have an opportunity to significantly reduce their GHG emissions. Therefore, facilities can respond to the growing climate emergency by not only building resilience to respond on the front lines of extreme weather events and long-term stresses and treat those made ill by the climate crisis, but also through the reduction and eventual elimination of all environmental contaminants released by their operations. Key areas for action include water, health care waste, sanitation and wastewater, chemicals, radiation, air quality and food (Box 1).

Box 1Climate resilience and environmental sustainability in relation to environmental determinants of health in health care facilities (HCFs).Water: Much of the health care delivery in developing countries still takes place in settings with inadequate or non-existent municipal water supply or water and wastewater treatment facilities, and in drought-prone areas made worse by climate change. HCFs need sufficient quantities of safe water to provide quality health care services [11]. Health care waste: Over half of the world’s population is estimated to be at risk from environmental, occupational or public health threats resulting from improperly treated health care waste [12]. Improper health care waste management can occur for several reasons, such as lack of awareness about the health hazards related to health care waste, inadequate training in proper waste management, lack of infrastructure or energy and lack of appropriate regulations or enforcement of existing regulations [13]. Sanitation and wastewater: In some settings, wastewater can be treated on site to remove chemicals that cannot be eliminated in municipal systems. In many countries, it is mandatory to reduce biological loading, and then treat the water in a municipal system. However, this is not alway possible in rural areas where no service is available or in cities where the municipality requires on-site treatment. In these situations, a range of affordable wastewater treatment technologies are available [14]. Chemicals: An estimated 1.6 million lives and 45 million disability-adjusted life years were lost in 2016 due to exposure to selected chemicals [15]. Chemicals are ubiquitous in HCFs and used for unique purposes, such as in chemotherapy to treat cancer, or as disinfectants for cleaning and sterilization. In addition, many medical devices such as thermometers, which contain mercury, are still in use [16]. Radiation: Direct patient exposure to ionizing radiation during medical procedures constitutes the largest anthropogenic source of population radiation exposure overall. Every year, an estimated seven million health workers are exposed to radiation due to their professional activities. While new health technologies, applications and equipment are rapidly being developed to improve the safety and efficacy of procedures, incorrect or inappropriate handling of these technologies may result in potential health hazards for patients, health workers and the general public [17]. Air quality: Ambient air pollution, which is principally driven by fossil fuel combustion, kills an estimated 4.2 million people annually [18]. Its health impacts, which include damage to the heart, lungs and every other vital organ, are exacerbated by climate change [19]. Many HCFs contribute to ambient air pollution through on-site fossil fuel energy combustion, medical waste incineration, the purchase of energy generated from fossil fuel sources and procurement of goods that are produced and transported using fossil fuels. Food: HCFs in many countries are major consumers of food and can therefore model and promote health and sustainability through their food choices. A growing number of HCFs in high-income and low- and middle-income countries that purchase and serve food to patients and workers are reducing their environmental footprint and improving patient and worker health by making changes in hospital service menus and practices. These include limiting the amount of meat in hospital meals, cutting out fast and junk food, composting food waste and producing their own food on site, as well as promoting sustainability by holding farmers’ markets for local producers to sell healthy food to the community, leading to community resilience [20].

### 3.1. Risks to Health Care Facilities from Climate Change

Climate threats to health systems are particularly disruptive for individuals and communities when they affect HCFs. Climate change can impact the delivery of health care services in large hospitals and small facilities, in high- and low-income settings alike. The increased frequency and intensity of many natural hazards challenges the infrastructure, support systems and supply chains that HCFs and their communities depend upon. For example, sea-level rise, or tropical storms with increased intensity, can cause increasingly widespread and prolonged flooding that disrupts vulnerable infrastructure and transportation systems, as well as the delivery of materials and food, and may lead to the release of hazardous substances, the contamination of the environment and health risks [21]. Often, HCFs are not built to physically and operationally cope with these and other climate-related risks, such as droughts, extreme temperatures, wildfires and changed patterns of climate-sensitive diseases. For example, in some countries, water scarcity and unpredictability in supply is increasingly affecting HCFs, preventing them from providing essential hygiene and infection prevention and control services. This is particularly important for facilities to respond to outbreaks.

All risks associated with climate change can impact the functioning of HCFs directly, and also result in increased demand for their services. For example, flooding can cause significant damage to hospital mechanical equipment while contaminating available water sources, and simultaneously the HCF must deal with an increased number of persons seeking health care. Prolonged high winds can damage roof-top equipment and cause structural damage to buildings and electric transmission lines and other public infrastructure. Health workers protect the health of their communities before, during and after disasters, as first responders to emergencies, but are also vulnerable to the impacts of extreme weather events.

### 3.2. Risks to Environmental Sustainability from Health Care Facility Operations

HCFs, when not well designed, equipped and managed, produce adverse environmental impacts, affecting the health of the health workforce and of the community they aim to serve and protect. A minimum requirement for climate resilient, safe and high-quality care is access to reliable sources of water and energy; yet many HCFs lack even these basic resources. Environmental sustainability, from this perspective, means implementing interventions that optimize the consumption of resources (such as water, energy, food), and reduce GHG emissions and waste discharge (including biological, chemical, radiological). It also includes procuring goods and services that follow the principles of environmental sustainability. Importantly, sustainability measures need to be evaluated for their performance and functionality, because quality of care should be the most important criteria. Therefore, more sustainable goods, materials and services should be considered when they do not compromise health care provision, and do not adversely affect the health and safety of health care workers.

Health care contributes to air pollution and GHG emissions through energy consumption (transport, electricity, heating and cooling) as well as product manufacture, use and disposal. Direct emission sources include those emanating directly from on-site fuel consumption in HCFs and vehicles owned by HCFs (known as Scope 1 GHG emissions). Indirect emissions refer to energy used by facilities, such as electricity, steam, cooling and heating (Scope 2 GHG emissions). A third significant source of emissions primarily derived from the health care supply chain is through the production, transport and disposal of goods and services, such as pharmaceuticals and other chemicals, food and agricultural products, medical devices and hospital equipment as well as instruments purchased and used by HCFs (Scope 3 GHG emissions) [22]. Several tools are available to measure GHG emissions, such as the Greenhouse Gas Protocol [23], and the IPCC guidelines for national GHG inventories [24].

### 3.3. The Policy Context

Of the global mandates for responding to climate resilient and environmentally sustainable health systems, the WHO Global Strategy on Health, Environment and Climate Change (the Strategy) and the 2030 Agenda for Sustainable Development are especially relevant for HCFs. Approved by the 72nd World Health Assembly in 2019, the Strategy covers all aspects of health and environment with emphasis on climate change and responds to health risks and challenges up to 2030. The Strategy has 12 goals, one of which responds to health care settings, with the goal that “All health care facilities and services are environmentally sustainable: using safely managed water and sanitation services and clean energy; sustainably managing their waste and procuring goods in a sustainable manner; are resilient to extreme weather events; and capable of protecting the health, safety and security of the health workforce” [25].

The 2030 Agenda for Sustainable Development, adopted by all UN Member States in 2015, provides a shared blueprint for peace and prosperity for the people and the planet, now and into the future [26]. At its heart are the 17 Sustainable Development Goals (SDGs), which are an urgent call for action by all countries in a global partnership. SDGs recognize that ending poverty and other deprivations must go hand in hand with strategies that improve health and education, reduce inequality and spur economic growth—all while tackling climate change. Making HCFs climate resilient and environmentally sustainable would contribute to achieving SDGs related to climate change, sustainable consumption, water and sanitation, energy, employment, resilient infrastructure and health and well-being (Table 1).

## 4. A Framework for Action

Climate resilient and environmentally sustainable HCFs can be defined as those able to anticipate, respond to, cope with, recover from and adapt to climate-related shocks and stresses, while minimizing negative impacts on the environment and leveraging opportunities to restore and improve it, so as to bring ongoing and sustained health care to their target population and protect the health and well-being of future generations [1,10] Here, we propose a framework for HCF action (Figure 3). There are three objectives under each of the four fundamental requirements to provide safe and high-quality care that are central to the action framework [1]. We propose this framework to develop a set of potential interventions that health sector decision makers can employ to enhance both climate resilience and environmental sustainability.

Many of the required interventions are linked and therefore may address multiple objectives that do not always fall neatly into one category. For example, a climate resilient intervention to cope with recurrent drought may be to harvest rainwater during the rainy season and store it in containers for later use. An environmentally sustainable intervention would be to ensure that the containers are properly sealed so that they do not become mosquito breeding sites and lead to vector-borne disease outbreaks. Building climate resilience and environmental sustainability are best addressed together for achieving synergies and resource efficiency.

### 4.1. Interventions to Strengthen Climate Resilience and Environmental Sustainability in Health Care Facilities

Given the complexity and variety of HCFs, there is no easy or simple way to identify all possible interventions required. Here, we propose beginning with the four fundamental priorities discussed earlier (health workforce; water, sanitation and health care waste; energy; and infrastructure, technology and products). Each of these can be subdivided into three objectives [1] for both climate resilience and environmental sustainability. This would produce 24 sets of interventions. Moreover, HCFs and health sector officials would need to reach out to decision makers outside of the health sector to collaborate on the implementation of some of the climate resilience or environmental sustainability measures. It is possible that HCF officials may not be able to complete all interventions at once, and that not all HCFs will have the required capacity and resources to undertake these interventions. Moreover, as it may not be possible to complete all of these interventions in a short period of time, it may be better to approach the use of this framework as supporting continuous improvement over time. Therefore, the identification of needed and priority interventions for a specific HCF will depend upon the local context. HCF staff would benefit by integrating information from this framework and the interventions into regular health facility planning processes, so as to be more efficient and maximize the use of resources. Interventions cannot be rated as either done or not done. We suggest a simple rating of interventions, into three performance groups: (i) low (indicates low performance, or unavailable activity, or unable to complete); (ii) medium (indicates medium performance, or activity in progress, or incomplete); and (iii) high (indicates high performance or completed activity, or achieved and tested).

### 4.2. Area 1: Health Workforce

Health workers have a key role in building climate resilience and environmental sustainability of HCFs. Health care workers are the main actors in ensuring that interventions are effective for their own roles and activities, as well as for other components of the framework. Because building climate resilience and environmental sustainability are relatively new approaches for health workers, building awareness, training and empowering health workers are key requirements for the successful implementation of interventions. In many settings, already stressed by lack of staff and resources, building resilience and environmental sustainability need to be integrated as a support to health care work and the protection of health and safety of staff, patients and communities. Objectives for the implementation of this component:Human resources—HCFs having a sufficient number of health workers with healthy and safe working conditions and capacity to deal with health risks from climate change, as well as the awareness and empowerment to ensure environmentally sustainable actions.Capacity development—training, information and knowledge management targeted at health care workers to respond to climate risks and minimize environmental threats resulting from the operation of the HCF.Communication and awareness raising—communicate, coordinate and increase awareness related to climate resilience and environmental sustainability among health workers, patients, visitors, target communities and with other sectors.

### 4.3. Interventions for Climate Resilience

The health workforce may be affected by two main mechanisms as a result of climate change. The first relates to changes in the frequency and intensity of extreme weather events and longer-term climatic changes that may affect the facility or the workforce’s ability to perform their duties, or even to reach the facility. The second is through changed patterns of climate-sensitive diseases, to which the health workforce may not have the experience to respond in a timely manner. One key requirement is, therefore, to have in place sufficient numbers of skilled and informed health workers. This is a constraint in many countries and needs urgent attention. Another key requirement is ensuring the health and safety of health workers, by identifying occupational hazards to prevent and control exposures (climate and non-climate related). The health workforce includes not only nurses and doctors and other health specialists, but also staff undertaking diverse occupational activities, such as administration, reception, radiography, maintenance, housekeeping, food service, laboratory work and specialist technicians and cleaning and laundry services, as well as hospital caretakers, dietitians, waste management staff, ambulance assistants and technicians and transport drivers. Workplace risks are therefore varied, and even those not in direct contact with patients can be exposed to contaminating agents of a biological, chemical, radiological or physical nature and to ergonomic and psychosocial hazards [27]. Efforts to achieve sustainable, healthy and safe working conditions in the health sector are thus important to both improving patient and community well-being, and for reducing the risks to workers engaged in health care-related activities. The interventions build overall resilience, and in particular climate resilience of the health workforce.

### 4.4. Interventions for Environmental Sustainability

Much of the environmental impacts from HCFs relate to issues associated with water, sanitation and hygiene (WASH), wastes (including biological, chemical and radiological hazards), energy and procurement practices. However, these are mediated in part by what the health workforce does or is unable to do. Thus, the health workforce has a large responsibility in ensuring environmentally sustainable practices through their actions. Health care waste, for example, is highly dependent on the actions of health workers. Impacts from biological, chemical or radioactive sources, if not adequately managed and disposed of, will affect health workers, as much as patients, visitors and surrounding communities. Therefore, this is an area particularly amenable to health workforce actions. However, even a well-informed health workforce may be unable to implement all needed actions in all areas. Water sources and energy sources, for example, may not depend on the HCF or its staff. Similarly, procurement may be done centrally, without consultation or inputs from specific HCFs. This implies that many interventions may need to be taken at levels other than the HCF itself and may depend on the local context, including the health system, actions by local governments and national policies.

### 4.5. Area 2: Water, Sanitation and Health Care Waste

The availability of sustainable water, sanitation and environmental, chemical and health care waste management services are essential to quality of care and infection prevention and control in HCFs. Important advances and commitments in this area have been achieved in recent years. The WHO, with international agencies, Member States and other partners are now actively responding to this critical component for health and well-being [28]. An example is WASH FIT, a tool to prioritize risks and make improvements [11]. These actions are also key to climate resilience and environmental sustainability as outlined in the proposed interventions below. The WHO Chemicals Road Map [16] requests action to prevent and mitigate chemical health risks in HCFs. Objectives for the implementation of this component are:Monitoring and assessment—information regarding water, sanitation, chemical use and health care waste management considers climate resilience and environmental sustainability for promoting action.Risk management—strengthened capacity of HCFs to manage water, sanitation, chemicals and health care waste risks to workers, patients and served communities, by including assessments of climate resilience and environmental sustainability in responding to hazards and identifying and reducing exposures and vulnerabilities.Health and safety regulation—water, sanitation, chemical safety and health care waste regulations are implemented, taking into consideration climate variability and change, and environmental sustainability.

### 4.6. Interventions for Climate Resilience

Lack of high-quality water, or irregular access, is a major problem in many HCFs in less developed regions, particularly in areas of natural water scarcity, and has implications for sanitation and hygiene. This problem is increasingly aggravated by climate variability and change and may result in a shortage of water for prolonged periods or excess water for short periods (drought followed by heavy rains and flash floods). Floods may also adversely impact sanitation systems and overflow of wastewaters. In periods of drought, and even when water becomes available, people may take different initiatives to overcome shortages, such as storing water, or accessing water of lower quality. Sea-level rise may also increase salinity in coastal aquifers, affecting water quality and flooding sewage systems. Achieving optimal use of water resources means that some HCFs may need to conserve water, while others need to increase their use. This needs careful consideration to ensure that actions in one area do not impact on other areas.

### 4.7. Interventions for Environmental Sustainability

Managing water, sanitation, chemical hazards and hazardous health care waste are essential components of an HCF’s environmental sustainability, in all countries and settings. According to the WHO, of the total amount of waste generated by health care activities, 15% is considered hazardous, which may be infectious, toxic or radioactive. Concerns include lack of proper disposal of syringes, open burning and incineration of health care wastes with consequent emissions of particulate matter (and in some cases of dioxins and furans, and toxic metals) and unintended release into the environment of pharmaceuticals, or chemical and biological hazards, including drug-resistant microorganisms [29]. Key areas of action include the substitution of harmful chemicals to improve the health and safety of patients, HCF staff, communities and the environment by using safer chemicals, materials, products and processes throughout HCFs. For the remaining waste (approximately 75–90%) generated in HCFs, which is considered non-hazardous, it is key to promote proper segregation and recycling. There is also a need to eliminate non-essential single-use plastics in health facilities and select plastic materials favoring reduced toxicity and the opportunity for reuse and/or recycling. Another concern is radiation safety. Direct patient exposure to ionizing radiations during medical procedures constitutes the largest anthropogenic source of population radiation exposure overall. Actions therefore include focus on enhancing safety and quality in the use of ionizing radiation in health facilities. Antimicrobial resistance (AMR) is a major global public health concern and a food safety issue. When pathogens become resistant to antimicrobial agents, they can pose a greater human health risk, resulting in potential treatment failure, loss of treatment options and increased likelihood and severity of disease. Problems related to AMR are inherently related to antimicrobial use in any environment, including human and non-human uses, and to wastewater disposal in HCFs.

### 4.8. Area 3: Energy

Access to electricity in HCFs is critical to achieving UHC and several SDG targets, including improving maternal health, reducing child mortality and preventing disease. Many HCFs, particularly those in rural areas, lack reliable, affordable electricity supply for powering basic services such as lighting, communications, refrigeration, diagnostics and the medical devices required for health services [30,31]. In addition, inefficient use of energy technologies, such as inefficient devices and appliances, contributes to fuel waste and costs and adds to air pollution. Objectives for the implementation of this component are:Monitoring and assessment—information regarding energy services should consider climate resilience and environmental sustainability for promoting action.Risk management—strengthened capacity of HCFs to manage energy-related risks to workers, patients and served communities, by including assessments of climate resilience and environmental sustainability in responding to hazards and identifying and reducing exposures and vulnerabilities.Health and safety regulation—regulations on energy use and access are implemented, taking into consideration climate variability and change, and environmental sustainability.

### 4.9. Interventions for Climate Resilience

Climate change can impact on energy access in many ways and in all types of HCFs. Although many HCFs lack regular electricity access (whether from an electricity grid or generated locally), climate change can further limit this access. Extreme weather events, such as storms, may destroy power lines or solar panels. Floods may affect generators or battery storage. Heat waves may increase electricity use in cities to power air conditioning, leading to rationing or outages. Most of these can be overcome with good planning for which an increasing number of resources are becoming available [32,33]. Nevertheless, an HCF can reduce its GHGs and become more resilient to electricity grid disruptions and unreliability. When on-grid energy is unavailable or unreliable, HCFs can develop and use off-grid systems. Solar energy can be harnessed through photovoltaic cells to heat water or generate electricity (which can be stored in batteries). District heating systems can create efficiency across cities and buildings, while closed-loop or low-enthalpy geothermal energy can provide a low-carbon thermal heating alternative. Energy can also be produced on site through other renewable sources, such as wind, biomass or hydroelectricity. The setting and scale of the HCF, as well as the availability of energy resources, can influence the selection of the most appropriate sustainable energy solution. Renewable energies can be deployed using both centralized and decentralized approaches. As renewable energies are clean, in both cases they contribute to environmental sustainability [34]. Renewable energies can be deployed on site with a decentralized approach in places already connected to the grid (such as facilities in urban areas) and those not connected to the grid (such as in rural areas). Decentralized renewable energy systems play a crucial role in climate resilience, such as during extreme weather events if the national grid gets damaged or on-site diesel generators have issues due to problems in the fuel supply chain.

### 4.10. Interventions for Environmental Sustainability

Much of the environmental and public health harm produced by energy consumption is from the combustion of fossil fuels, such as oil, coal and gas. GHG emissions and air pollution generated from fossil fuel combustion are major contributors to global climate change and local health problems. By increasing energy efficiency and transitioning to clean, renewable energy sources, the health sector can reduce GHG emissions and contribute to protecting public health from the impacts of climate change and air pollution. Indirectly, these changes may bring with them the health and economic co-benefits of reductions in hospital admissions and treatments for chronic illnesses such as asthma and lung and heart disease caused by the pollution created from the extraction, refining and combustion of coal, oil and gas. HCFs can promote energy conservation and efficiency and implement renewable energy strategies and procurement, reducing GHG emissions and saving financial resources, while maintaining or improving care quality. Key areas requiring action include:Building characteristics: the quality of the building and its features affect the energy demand through the quality of insulation of walls and windows, the use of passive cooling and shading options and its location and exposure to climate and weather.Energy efficiency: electric lighting fixtures can consume a large proportion of electrical energy and, depending upon the source, can contribute to internal heat loads. Efficient appliances and thermal insulation also contribute to energy efficiency.Transportation: transportation is a major source of both air pollution and GHG emissions, and the health sector—with its fleet of ambulances, hospital vehicles and delivery vehicles, as well as staff and patient travel—is a transportation-intensive industry.Food: purchased, prepared and provided in a variety of health care settings contributes to GHG emissions of the health care sector.Pharmaceuticals: produced in facilities that use a great deal of energy and emit significant GHGs. While selecting and prescribing medicines, it may be possible to consider those manufactured with the least environmental impact.

### 4.11. Area 4: Infrastructure, Technology and Products

There are both structural and non-structural elements of concern. Structural elements are those that form part of the load-bearing system of the building, such as columns, beams, walls, floor slabs and foundations. Structural measures for HCFs would include construction built to resist floods, storms or sea-level rise. Non-structural elements are those critical to the functioning of the HCFs. These include the architectural elements, emergency access and exit routes to and from the HCF, critical systems (such as electricity, water supply, waste management, fire protection), medical, laboratory and office equipment (whether fixed or mobile) and supplies used for analysis and treatment [35], as well as emerging technologies with important recent advances globally (such as digital health, including tracking of disease outbreaks). Non-structural measures also include awareness raising, training and education [36], which are included under the health workforce. Objectives for the implementation of this component are:Adaptation of current systems and infrastructures—building regulations implemented in the construction and retrofitting of HCFs to ensure climate resilience and environmental sustainability.Promotion of new systems and technologies—adopt new technologies and processes that can provide climate resilience, environmental sustainability and enhanced health service delivery.Sustainability of health care facility operations—adopt and procure low environmental impact technologies, processes and products to enhance climate resilience and environmental sustainability.

### 4.12. Interventions for Climate Resilience

The structural and non-structural components and measures when fully functional would help HCFs to remain operational during and after shocks or stress to protect the health of their communities. Components also include construction materials, which should not result in occupational or environmental hazards. Measures include climate resilience of essential environmental services to health facilities, such as water and sanitation services, chemical safety and electricity and energy services, which may be compromised by climate variability and change [1], as well as preparations in terms of procurement and logistics. Because of their importance, water and sanitation, chemical safety and energy services are addressed separately.

### 4.13. Interventions for Environmental Sustainability

Low environmental impact technologies, processes and products need to be adopted for environmental sustainability. One key component is the procurement of goods and services. A sustainable procurement program aims to reduce carbon emissions and chemical pollution, and conserve natural resources by identifying environmentally sustainable goods and services with fewer harmful effects on human health and the environment. The health care sector consumes vast amounts of natural and processed resources that are sourced, manufactured and delivered across the sector’s supply chain. The production, transport and disposal of health care goods and services, such as pharmaceuticals and chemicals, food and agricultural products, medical devices, hospital equipment and instruments all make a substantial contribution to the sector’s environmental footprint, including generation of waste, chemical contamination and GHG emissions. Table 2 presents examples of interventions.

## 5. Conclusions

HCFs are impacted by climate change and also by their surrounding environments. In turn, in their operations, HCFs produce GHGs, contributing to climate change, and through their emissions and unsustainable practices, they may contribute to environmental degradation. This guidance aims to ensure ongoing climate resilience and environmental sustainability of HCFs. It builds on the premise that HCFs need to satisfy a minimum of four requirements to provide safe and high-quality care: (i) a skilled and informed workforce, (ii) adequate water, sanitation and waste services, (iii) energy services and (iv) safe, functional and sustainable infrastructure, including technologies and products. Simultaneously, HCFs need to consider how these four basic requirements can be strengthened to become climate resilient and how they can contribute to environmental sustainability.

It is important that these concepts are adapted to local realities and needs, building on local skills and knowledge. New advances in knowledge, experiences and lessons from several HCFs and changed circumstances (such as those brought about by public health emergencies like the COVID-19 pandemic) imply that any set of proposed interventions must be used with flexibility, and more as a model on how to improve operations, than as a prescription with expected actions. Whether large or small, all HCFs can improve their operations while addressing key environmental concerns.

## Figures and Tables

**Figure 1 ijerph-17-08849-f001:**
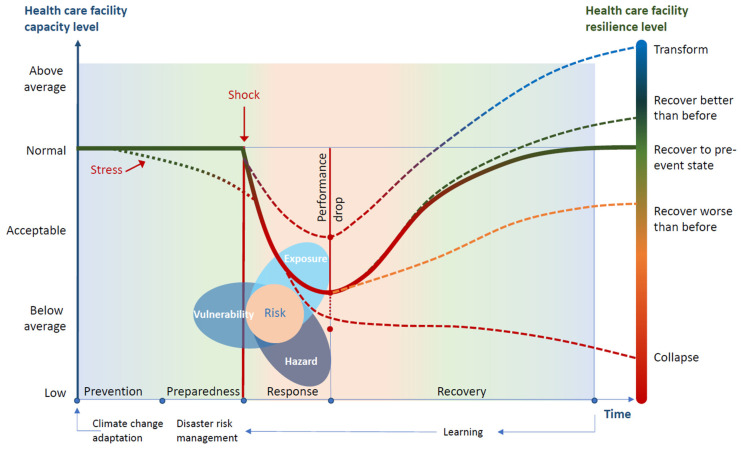
Climate resilience in health care facilities. Sources: [6,7,8,9].

**Figure 2 ijerph-17-08849-f002:**
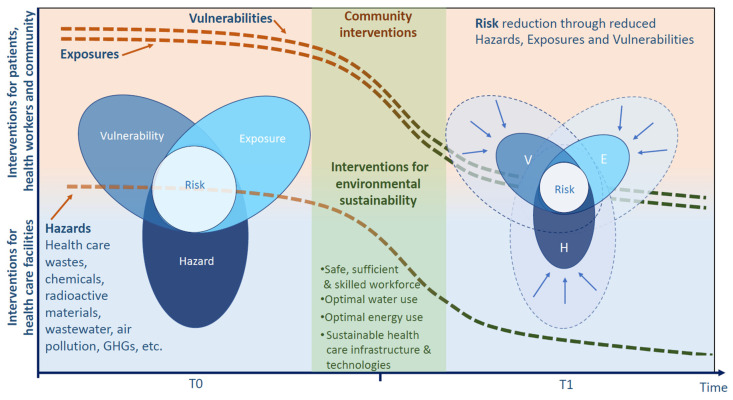
Environmental sustainability in health care facilities.

**Figure 3 ijerph-17-08849-f003:**
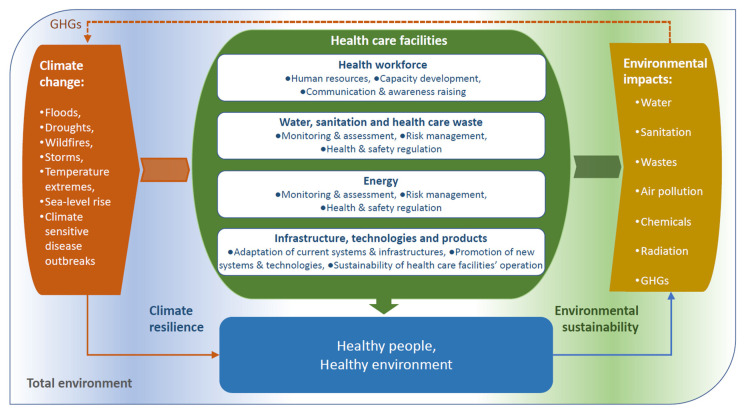
Framework for building climate resilient and environmentally sustainable health care facilities.

**Table 1 ijerph-17-08849-t001:** Selected Sustainable Development Goals (SDGs) and targets with implications for health care facilities.

SDGs	Targets	HCF Action Areas
13. Take urgent action to combat climate change and its impacts	13.1 Strengthen resilience and adaptive capacity to climate-related hazards and natural disasters in all countries	Health workforce; Infrastructure, technologies and products
	13.2 Integrate climate change measures into national policies, strategies and planning	Health workforce; Infrastructure, technologies and products
	13.3 Improve education, awareness-raising and human and institutional capacity on climate change mitigation, adaptation, impact reduction and early warning	Health workforce
12. Ensure sustainable consumption and production patterns	12.4 By 2020, achieve the environmentally sound management of chemicals and all wastes throughout their life cycle, in accordance with agreed international frameworks, and significantly reduce their release to air, water and soil in order to minimize their adverse impacts on human health and the environment	Water, sanitation and health care waste
	12.5 By 2030, substantially reduce waste generation through prevention, reduction, recycling and reuse	Water, sanitation and health care waste
	12.7 Promote public procurement practices that are sustainable, in accordance with national policies and priorities	Infrastructure, technologies and products
6. Ensure availability and sustainable management of water and sanitation for all	6.1 By 2030, achieve universal and equitable access to safe and affordable drinking water for all	Water, sanitation and health care waste
	6.3 By 2030, improve water quality by reducing pollution, eliminating dumping and minimizing release of hazardous chemicals and materials, halving the proportion of untreated wastewater and substantially increasing recycling and safe reuse globally	Water, sanitation and health care waste
	6.4 By 2030, substantially increase water-use efficiency across all sectors and ensure sustainable withdrawals and supply of freshwater to address water scarcity and substantially reduce the number of people suffering from water scarcity	Water, sanitation and health care waste
7. Ensure access to affordable, reliable, sustainable and modern energy for all	7.1 By 2030, ensure universal access to affordable, reliable and modern energy services	Energy
	7.2 By 2030, increase substantially the share of renewable energy in the global energy mix	Energy
	7.3 By 2030, double the global rate of improvement in energy efficiency	Energy
8. Promote sustained, inclusive and sustainable economic growth, full and productive employment and decent work for all	8.8 Protect labour rights and promote safe and secure working environments for all workers, including migrant workers, in particular women migrants, and those in precarious employment	Health workforce
9. Build resilient infrastructure, promote inclusive and sustainable industrialization and foster innovation	9.4 By 2030, upgrade infrastructure and retrofit industries to make them sustainable, with increased resource-use efficiency and greater adoption of clean and environmentally sound technologies and industrial processes, with all countries taking action in accordance with their respective capabilities	Infrastructure, technologies and products
3. Ensure healthy lives and promote well-being for all at all ages	3.8 Achieve universal health coverage, including financial risk protection, access to quality essential health-care services and access to safe, effective, quality and affordable essential medicines and vaccines for all	Access to health care facilities
	3.9 By 2030, substantially reduce the number of deaths and illnesses from hazardous chemicals and air, water and soil pollution and contamination	Water, sanitation and health care waste

Source: [26].

**Table 2 ijerph-17-08849-t002:** Sample interventions.

	Objectives	Climate Resilience	Environmental Sustainability
Health workforce	Human resources	Identify minimum needs in terms of health care workers to ensure the operational sufficiency of every HCF department, in case of climate-related disaster or emergency	Increase human resources available to reduce or eliminate disease burden among vulnerable populations resulting from environmental hazards in HCFs
	Capacity development	Health workforce receives training and exercises for preparing for, responding to and recovering from extreme weather-related emergencies	Education and training provided to HCF staff and the community on environmental factors that contribute to the burden of disease
	Communication and awareness raising	Key messages for target audiences (such as patients, staff, public) drafted in preparation for the most likely extreme weather disaster scenarios	Increase knowledge and communication about the environmental impact of pharmaceuticals and their disposal
Water, sanitation and health care wastes	Monitoring and assessment	Develop climate resilient water safety plans	Implement and monitor a waste reduction program including waste management training for all staff
	Risk management	WASH climate risk management plan implemented	Wastewater is safely managed through use of on-site treatment or sent to a functioning sewer system
	Health and safety regulation	Sanitation technologies designed to be more resistant to climate hazards and able to operate under a range of climate conditions	Harvested rainwater or gray water is safely used to flush toilets, clean outdoor pavement areas and water plants when possible
Energy	Monitoring and assessment	Assess that location of energy backup or renewable energy infrastructure can withstand extreme weather events (such as strong winds, hail, floods)	Assess the HCF to determine how and where energy use can be reduced (or increased in energy poor areas)
	Risk management	Plan developed for managing intermittent energy supplies or system failure	HCF fossil fuel consumption reduced by use of renewable energy sources, including solar (photovoltaic) power, wind power, hydro power and biofuels
	Health and safety regulation	Adequate lighting, communications, refrigeration and sterilization equipment are available during climate-related disasters or emergencies	Developed an energy management plan to measure energy consumption
Infrastructure, technology. Products	Adaptation of current systems and infrastructures	HCFs built or retrofitted to cope with extreme weather events, ensuring their resilience, safety and continuous operation	New (or retrofitted) HCFs designed and constructed based on low-carbon approaches
	Promotion of new systems and technologies	HCF uses proven smart materials and applications, sensors, low-power electronics, telemedicine and similar health care-appropriate technology	Substitute mercury-containing thermometers and blood pressure-measuring devices for affordable, validated device alternatives
	Sustainability of HCF operations	Anticipate the impact of the most likely disaster events on the supply of water, food and energy	Implement a clear environmentally sustainable procurement policy statement or protocol for all types of products, equipment and medical devices used in the HCF

N.B. The complete set is available in the WHO’s guidance for climate resilient and environmentally sustainable heath care facilities [37].

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
