# Peer review of "Towards Climate Resilient and Environmentally Sustainable Health Care Facilities"

_ijerph, 2020, doi:10.3390/ijerph17238849_

Round 1

Reviewer 1 Report

The paper entitled “Towards climate resilient and environmentally sustainable health care facilities” seems focusing on the environmental issues relating to safe and quality health care facilities. I found lack of cohesion in the manuscript. The description is difficult to understand for the general readers. The manuscript needs to be revised properly for the possible publication. However, I would like to provide the following comments.

  1. I suggest the author to focus on word selection. In line 25, “the release of pollution….” The word ‘pollutants’ is appropriate than ‘pollution’.
  2. In many lines, I have found that the fictional description. I suggest to provide references for lines 43-44, lines 57-59, Box 1 third paragraph and lines 398-399.
  3. In lines 105-106, reference no 8 associated with the description related to figure 1. But, the reference is not mentioned in sources.
  4. The description given in lines 167-169 is not clear. Please, clarify the statement.
  5. In figure 3, Climate change section, the word ‘drought’ is repeated.
  6. In the manuscript, the author has pointed the problems of HCFs associated to environment. But, the suggestions to overcome form the problems is lacking. The ides to control environmental pollution is more important. Many areas like Energy consumption, Building Efficiency, Thermal insulation are included. The discussion is insufficient.
  7. Different radioactive isotopes are used during diagnosis and treatment. The ideas to tackle and dispose of these radioactive isotopes need to be discussed.
  8. The local skill, local knowledge must be addressed for the improvement of HCFs.
  9. In conclusion, the ideas of author is required.  

Author Response

We thank the reviewer for these comments. We have addressed their concerns as follows:

  • Line 25, changed from “pollution” to “pollutants” as requested
  • References to lines 43-44 and 57-59 were not added because they are not required. It is the statement of the authors. Box 1, 3rdparagraph has a reference. Lines 398-399, we added a missing reference.
  • Lines 105-106. The missing reference was added
  • Description in lines 167-169, text was added to clarify (please see track changes)
  • Repeated word in figure 3. The figure was corrected.
  • We have not added text related to the different types of radioactive isotopes. That is too specific for this paper.
  • Local skills and local knowledge. Text was added on this (please see track changes).
  • Regarding the overall style of the paper, we also followed the advice of other reviewers to improve readability where needed.

Reviewer 2 Report

The text presents the problem of climate resilience and sustainability of health care facilities (HCF). The authors define the background of the problem and outline basic issues connected with climate change and HCFs. They present a framework for building climate-resilient and environmentally sustainable health care facilities, which is their input to the discussion. 

The manuscript is well prepared and the content is presented in a clear manner. The change I suggest is to emphasize in the abstract that the authors propose a framework for building climate-resilient and environmentally sustainable health care facilities.

line 145: authors write that HCFs „have an opportunity to significantly reduce global GHG emissions”. It would be good to check what is the HCFs share in global GHG emission - it is probably not so significant

line 175: there is: well designed, there should be: well-designed

area 3: authors could include ISO 5000 (energy management)

Author Response

We thank the reviewer for these comments and supportive comments. We have addressed their concerns as follows:

  • We have added text regarding the framework in the Abstract (please see track changes).
  • We do not have information on the fraction of global GHG emissions from HCFs. Different estimates are available for the whole health system in countries. Some studies point to around 4.5% of the total. We modified the text to avoid comparison with global emissions. However small, each HCF can reduce their Carbon footprint and contribute to the overall aim of reducing emissions.
  • Line 175. We made the correction requested (please see track changes).
  • We did not include information on ISO 5000 as this is too detailed for the purpose of this paper.

Reviewer 3 Report

Health Care Facilities are a well-chosen critical site for illustrating the complex intersection of scientific and social variables around risk, climate change, and conditions for stakeholder engagement.  I appreciate the overview of how the complex variables intersect in what are usually seen in Canada as too often insurmountably separated CIHR, NSERC and SSHRC concerns and styles of presentation.  The paper lays out the illustrative quality of the choices to fit its topic and the scope of this journal.  I am strongly recommending publications because I am certain those these conventional silos must be transcended.  The consistency of messages from WHO and other health professionals underscores the urgency of COVID-19 response now, while making it clear that other urgent challenges like natural disasters engage the same complex intersection of variables.

Ethical issues are rarely black and white.  The careful laying out of positive and negative sides of various chosen interventions outlines the kind of balance necessary for actors from all sides of science, society and public policy to communicate effectively and take stands for the long haul.  I hope that readers from very different backgrounds will take away better appreciation that they need others with very different skills to address this complexity with mutual respect. 

There is potential generalizability in the clarity with which the paper focuses on illustrative interventions, leaving readers with other foregrounded issues to apply the same methodologies to their own need for interventions.  We can't do everything simultaneously but we can choose a point of attack and take effective action one step at a time, in appreciation of variable local conditions and priorities.  

Author Response

We thank the reviewer for the positive review and support to this work.

Round 2

Reviewer 1 Report

The paper entitled “Towards climate resilient and environmentally sustainable health care facilities” seems focusing on the environmental issues relating to safe and quality health care facilities. The revised version seems improved. But, I also suggest to address all the comments.

  1. In the manuscript, the author has pointed the problems of HCFs associated to environment. But, the suggestions to overcome form the problems is lacking. The ideas to control environmental pollution is more important. Many areas like Energy consumption, Building Efficiency, Thermal insulation are included. The discussion is insufficient.
  2. In conclusion, the ideas of author is required.  

Author Response

We thank the reviewer for their comments.

The reviewer states: “In the manuscript, the author has pointed the problems of HCFs associated to environment. But, the suggestions to overcome form the problems is lacking. The ideas to control environmental pollution is more important. Many areas like Energy consumption, Building Efficiency, Thermal insulation are included. The discussion is insufficient”. Then the reviewer comments: “In conclusion, the ideas of author is required”.

The paper aims to address two separate but linked issues: climate resilience and environmental sustainability, as they affect health care facilities. We define this in lines 56-59. This becomes the framework for action, starting in line 228. Then, starting in line 249, we state “Given the complexity and variety of HCFs, there is no easy or simple way to identify all possible interventions required. Here we propose beginning with the four fundamental priorities discussed earlier (health workforce; water, sanitation and health care waste; energy; and infrastructure, technology and products)”. Under each of these four areas, we discuss “Interventions for climate resilience” and “interventions for environmental sustainability”. Specifically, for health workforce, we explain interventions for climate resilience in lines 287-306; for environmental sustainability in lines 307-320. Under water, sanitation and health care waste, we discuss interventions for climate resilience in lines 341-352. For environmental sustainability in lines 353-375. Under energy we address interventions for climate resilience in lines 392-413; and for environmental sustainability in lines 414-440. Finally, under infrastructure, technology and products we discuss interventions for climate resilience in lines 462-470. For environmental sustainability in lines 471-482.

In addition to the above, we include a large table, starting in line 485, listing 24 sample interventions in climate resilience and environmental sustainability. Moreover, at the bottom of that table we guide the reader to a full set of interventions (line 486).

It is, therefore, our opinion that there is not lack of suggestions to overcome the problems. We trust that the reviewer and editor will agree that adding further suggestions of interventions, would not add to the completeness of the paper.